



# On the origin of the mesospheric quasi-stationary planetary waves in the unusual Arctic winter 2015/16

Vivien Matthias[1] and Manfred Ern[2]

[1]Leibniz-Institute of Atmospheric Physics, Schloss-Str. 6, 18225 Kühlungsborn, Germany
[2]Institut für Energie- und Klimaforschung, Stratosphäre (IEK-7), Forschungszentrum Jülich GmbH, 52425 Jülich, Germany
*Correspondence to:* Vivien Matthias (matthias@iap-kborn.de)

**Abstract.** The mid winter 2015/16 was characterized by an unusually strong polar night jet (PNJ) and by extraordinarily large stationary planetary wave (SPW) amplitudes in the subtropical mesosphere. The aim of this study is to find the origin of these mesospheric SPWs in mid winter 2015/16. The time period studied here is split into two periods. The first period runs from late December 2015 until early January 2016 and the second period from early January until mid January 2016. While the SPW 1 dominates in the subtropical mesosphere in Period I, it is the SPW 2 that dominates in Period II. There are three possibilities how SPWs can occur in the mesosphere: 1) they propagate upward from the stratosphere, 2) they are in situ generated by longitudinally variable gravity wave (GW) drag, or 3) they are in situ generated by barotropic and/or baroclinic instabilities. Using global satellite observations from the Microwave Limb Sounder (MLS) and from the Sounding of the Atmosphere using Broadband Emission Radiometry (SABER) the origin of the mesospheric SPWs is investigated for both time periods. We found that due to the strong PNJ the SPWs were not able to propagate upward into the mesosphere northward of 50°N but were deflected upward and equatorward into the subtropical mesosphere. We show that the SPWs observed in the subtropical mesosphere are the same SPWs as in the mid-latitudinal stratosphere. At the same time we found evidence that the mesospheric SPWs in polar latitudes were in situ generated by longitudinally variable GW drag and that there is a mixture of in situ generation by longitudinally variable GW drag and by instabilities in mid latitudes. Our results based on observations show that every three mechanisms, upward propagating SPW and in situ generated SPWs by longitudinally variable GW drag and instabilities can act at the same time which confirms earlier model studies. Additionally, a possible contribution or impact of the unusually strong SPWs in the subtropical mesosphere to the disruption of the QBO in the same winter is discussed.

## 1  Introduction

The Arctic winter 2015/16 of the middle atmosphere was extraordinary in many respects. Firstly, in early winter the polar vortex was the coldest and strongest for the last 68 years (Matthias et al., 2016) with zonal mean wind speeds of over 80 ms$^{-1}$ around the stratopause in mid-latitudes. Secondly, a significant disruption of the quasi-biennial oscillation (QBO) occurred, beginning at the end of December 2015 and fully completed by mid-April 2016 (Osprey et al., 2016; Newman et al., 2016; Coy et al., 2017). Thirdly, this winter was also characterized by one of the strongest El Niño events on record with a strong polar stratospheric signature (Palmeiro et al., 2017). Additional to these global anomalies, there was a regional reversal in zonal





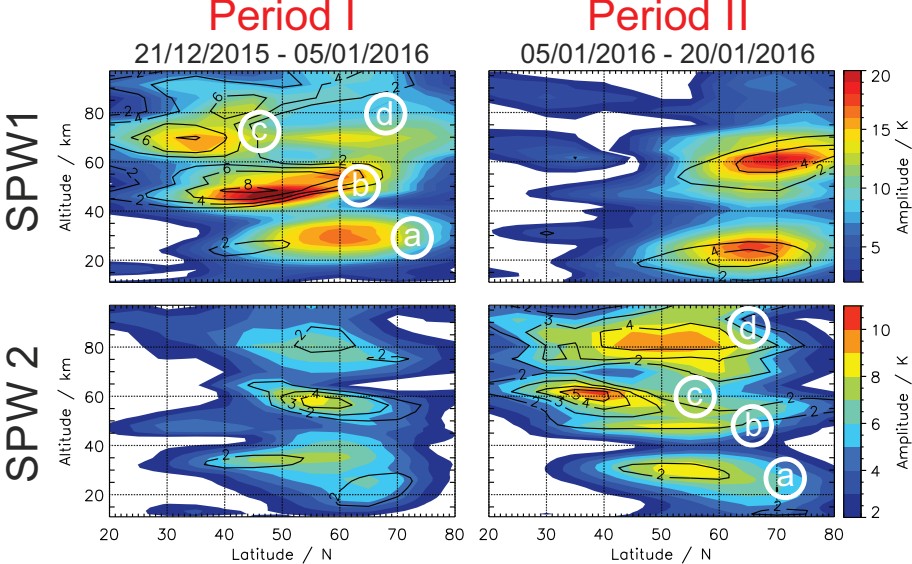

**Figure 1.** Latitude-altitude cross-section of the SPW 1 and 2 derived from MLS Temperature data. The amplitude as it occurred in the Arctic winter 2015/16 is color coded, the deviation from the 12-year mean is presented in contour lines for the respective time period from MLS data. The first half of the time period investigated here is called hereafter Period I and the second half Period II.

wind from eastward to westward in the mesosphere over an altitude range of $\sim$10 km starting at the end of December 2015 and lasting for four weeks which was not connected to a sudden stratospheric warming (SSW) (Stober et al., 2017). Stober et al. (2017) found out that this wind reversal, located only in polar latitudes, was caused by an unusual large stationary planetary wave (SPW) amplitude in the subtropical mesosphere leading to a changed residual meridional circulation. The main aim of

this paper is to find the origin of the significantly enhanced SPW amplitude in this time period in the subtropical mesosphere.

    The time period investigated here (December 21 2015 to January 20 2016) can be split into two periods. While the amplitude of the SPW 1 dominates in the first half of the time period in the subtropical mesosphere, the amplitude of the SPW 2 is strongly increased in the second half (see Fig. 1). Thus we call hereafter the first half of the period where the SPW 1 dominates Period I, and the second half where the SPW 2 dominates Period II.

In Fig. 1 the latitude-altitude cross-sections of the SPW 1 and 2 amplitude show four maxima in each period denoted with the letters $a$ to $d$. In Period I, maximum $a$ of the SPW 1 amplitude is located in the stratosphere between 40° and 75°N. Maximum $b$ is centered around the stratopause region and extends from 30° to 70°N while maximum $c$ is located between 60 and 80 km and between 20° and 50°N. This maximum $c$ is at least twice as large as the 12-year mean for this time period and beyond the standard deviation (not shown). Maximum $d$ also extends between 60 and 80 km but between 50° and 80°N.

In Period II, maximum $a$ of the SPW 2 amplitude is located in the stratosphere between 40° and 70°N. Maximum $b$ extends between 40 and 50 km from 35° to 65°N while maximum $c$ is located between 50 and 70 km and between 25° and 50°N. This





maximum $c$ is again at least twice as large as the 12-year mean for this time period, and beyond the standard deviation (not shown). Maximum $d$ is located between 60 and 80 km and between $30°$ and $70°$N.

While the origin of the maxima $d$ in the middle and polar latitudes in both periods will play a secondary role in this study, the main focus is on the origin of the maxima $c$ in the subtropical mesosphere.

There are three possible mechanisms for the occurrence of SPWs in the mesosphere: 1) the stratospheric SPWs propagate vertically into the upper mesosphere, 2) the SPWs are in situ generated by breaking or dissipation of gravity waves (GWs) that were filtered longitudinally variable in the stratosphere by SPWs (e.g., Holton, 1984; Smith, 2003; Lieberman et al., 2013) or 3) by barotropic and/or baroclinic instabilities (e.g., Siskind et al., 2010). Smith (1997) found that the two first mechanisms operate

but which one dominates depends on how favorable are the conditions in the middle atmosphere for vertical propagation. The third mechanism is well-known for the in situ generation for example of the quasi 2-day wave (e.g., Ern et al., 2013). A model study by Smith (2003) showed that vertical propagation of SPW dominates up to the lower mesosphere while in situ GW-generated SPWs dominate in the upper mesosphere. This result was confirmed by Lieberman et al. (2013) using satellite observations. In a model study, Siskind et al. (2010) discussed a possible in situ generation of SPWs in the mesosphere by

a combination of forcing from below combined with in situ instability. The in situ generation of SPWs by instabilities in the lower mesosphere was confirmed by Iida et al. (2014) using satellite observations.

The planetary wave guide (PWG) of SPWs are regions in the atmosphere where the background zonal wind supports the upward propagation of PWs (Dickinson, 1968). In midwinter the PWG is commonly split into a southern and a northern channel (e.g., Dickinson, 1968; Chapman and Miles, 1981; Li et al., 2007). While in the southern channel the SPWs preferably

propagate equatorward towards the subtropical zero wind line and barely reach the upper stratosphere (Albers et al., 2013), the northern PWG channel follows the polar night jet (PNJ) sometimes reaching even up into the mesosphere (Lin, 1982). The vertical propagation of SPWs depends on the strength and structure of the zonal mean zonal wind in the middle atmosphere (Lin, 1982).

We assume that the unusually strong PNJ in winter 2015/16 guided the SPWs towards the subtropical mesosphere and that

the conditions were favorable there for vertical propagation into the mesosphere.

In this paper we want to retrace the origin of the SPWs occurring in the subtropical mesosphere between December 21 2015 and January 20 2016. Therefore we use different diagnostic tools applied to global satellite data mainly from the Microwave Limb Sounder (MLS), as described in section 2. The propagation properties of the SPWs are shown and discussed in sections 3 and 4 for Period I and II, respectively. The cause for the change in the wavenumber between Period I and II is discussed in

section 5 and followed by a short discussion on the origin of the polar mesospheric SPWs in section 6. Finally, the results of this study are summarized in section 7.





## 2   Instruments and Methods

To find the origin of the quasi-SPW (hereafter SPW) in each period we need to know the characteristics of the SPW (wave-number and propagation direction) as well as the conditions for propagation (zonal wind and refractive index squared) in all latitudes and altitudes of the northern hemisphere.

Since this is a pure observational study, we use global temperature and geopotential height (GPH) data from the Microwave Limb Sounder (MLS) on board the Aura satellite (Waters et al., 2006; Livesey et al., 2015). MLS has a global coverage from 82°S to 82°N on each orbit, and a usable height range from approximately 11 to 97 km (261 – 0.001 hPa) with a vertical resolution of ∼4 km in the stratosphere and ∼14 km at the mesopause. The temporal resolution is 1 day at each location, and data are available since August 2004 until today (Livesey et al., 2015). Version 4 MLS data were used and the most recent
recommended quality screening procedures of Livesey et al. (2015) have been applied.

For our analyses the original orbital MLS data are accumulated in grid boxes with 20° grid spacing in longitude and 5° in latitude. Afterwards they are averaged at every grid box and for every day, generally resulting in a global grid with values at every grid point.

The estimation of the amplitude and phase of the SPWs as well as the filtering of those waves in atmospheric parameter is
done by using the two-dimensional least squares method of Wu et al. (1995).

To estimate the propagation condition for SPWs and the propagation direction of the SPWs itself, the zonal and meridional wind are needed. From the GPH data of MLS we calculate the geostrophic zonal ($u_g$) and meridional ($v_g$) wind by

$$u_g = -\frac{1}{f}\frac{\partial \Phi}{\partial y} \qquad v_g = \frac{1}{f}\frac{\partial \Phi}{\partial x} \tag{1}$$

where $\Phi$ is the geopotential, $f$ is the Coriolis parameter, and $x$ and $y$ are used to denote the partial derivatives $(a\cos\phi)^{-1}\frac{\partial}{\partial \lambda}$
and $a^{-1}\frac{1}{\partial \phi}$ where $\lambda$ is longitude, $\phi$ is latitude and $a$ is the radius of the earth.

A useful tool to distinguish regions of wave propagation from wave evanescence is the refractive index squared $n^2$. Here we use the spherical form of the quasi-geostrophic refractive index squared (Andrews et al., 1987):

$$n^2(\phi,z) = \frac{\bar{q}_\phi}{\bar{u}_g - c} - \left(\frac{s}{a\cos\phi}\right)^2 - \left(\frac{f}{2NH}\right)^2, \tag{2}$$

where

$$\bar{q}_\phi = \frac{2\Omega\cos\phi}{a} - \frac{1}{a^2}\left(\frac{(\bar{u}_g\cos\phi)_\phi}{\cos\phi}\right)_\phi - \frac{f^2}{\rho}\left(\rho\frac{(\bar{u}_g)_z}{N^2}\right)_z \tag{3}$$

is the zonal mean potential vorticity gradient, $z$ is the height, $s$ the spherical wavenumber, $c$ the phase velocity of the wave, $N(z)$ is the buoyancy frequency, $H$ is the scale height (=7 km), $\rho = \rho_0\exp(-z/H)$ is the standard density in log-pressure coordinates, $\Omega$ is the Earth's rotation frequency, overbars denote zonal mean quantities and subscripts denote derivatives with respect to the given variable. Planetary waves can propagate in regions where $n^2 > 0$ and are evanescent in regions where
$n^2 < 0$.

The direction and strength of SPW propagation is measured by the Eliassen-Palm flux (EPF) vectors and their divergence (EPFD) which is locally parallel to the group velocity of SPW (Edmon et al., 1980). The quasi-geostrophic form of the EPF





vectors $(\boldsymbol{F})$ and its divergence $(\nabla \cdot \boldsymbol{F})$ is defined as (e.g., Andrews et al., 1987):

$$\boldsymbol{F} = \left(F^{\phi}, F^{z}\right) = \rho a \cos \phi \left(-\overline{v_g' u_g'}, f \frac{\overline{v_g' \theta'}}{\theta_z}\right) \tag{4}$$

$$\nabla \cdot \boldsymbol{F} = \left(\frac{1}{\rho a \cos \phi}\right) \left(\frac{1}{a \cos \phi} \left(F^{\phi} \cos \phi\right)_{\phi} + F_z^z\right) \tag{5}$$

where $\theta$ is the potential temperature and primes denote the perturbation from the zonal mean. Note, that an interaction of a

SPW with the mean flow would result in EPFD convergence (negative) or divergence (positive) which makes a westerly wind deceleration or acceleration respectively (Andrews et al., 1987).

Since it is well known, that an upward propagating SPW has a westward tilt with height (e.g., Smith, 2003) we are interested in the phase tilt with height in each latitude. We estimate the phase behavior with height by filtering the geostrophic zonal wind for the wavenumber of interest, choosing a minimum in the lower stratosphere and follow it up into the mesosphere. If this

minimum ends at -180°E before reaching the uppermost level the longitude-altitude cross-section is copied and put to the left so that the minimum can be followed further upward.

To investigate the possible in situ generation of SPWs in the mesosphere by dissipating GWs longitudinally filtered in the stratosphere by SPWs, the GW absolute drag is calculated from SABER data. The SABER instrument, short for Sounding of the Atmosphere using Broadband Emission Radiometry, was launched onboard the TIMED satellite measuring temperatures

from 10 to 115 km (Mlynczak, 1997; Russell III et al., 1999; Yee et al., 2003). SABER switches between southward-viewing (83°S – 50°N) and northward-viewing (50°S – 83°N) geometries every about 60 days for about 60 days. For the period of interest, until early January 2016 SABER was in the southward-viewing part of the yaw cycle and switched to northward-view on 4 January 2016. Thus we can examine a possible in situ generation of polar mesospheric SPW by longitudinally variable GW drag for the second half of the time period investigated here only. The GW drag is calculated via a multi step procedure as

described in Ern et al. (2016) and in more detail in Ern et al. (2011). The end product is interpolated on a horizontal grid with a grid resolution of 10° in longitude and 2° in latitude. The vertical resolution is 10 km ranging from 30 to 90 km.

Using the synergy of the above-described analysis methods and satellite data sets, the origin of the mesospheric SPWs in each period is investigated in the following sections.

## 3  The origin of the subtropical mesospheric SPW 1 in Period I

Figure 2a shows the latitude-altitude cross-section of the zonal mean zonal wind (colored contour) and its deviation from the 12-year mean (contour lines). The PNJ is up to 25 ms$^{-1}$ stronger in Period I than in the 12-year mean. These stronger winds range from the mid-latitude stratosphere up into the subtropical mesosphere with the magnitude of the enhancement gradually decreasing towards the subtropical mesosphere.

The amplitude distribution of the SPW 1 matches the region of increased zonal wind (see Fig. 2b), i.e. the area of increased

amplitudes shifts southward with height. This shift is in accordance with the area in which the SPW 1 can not propagate due



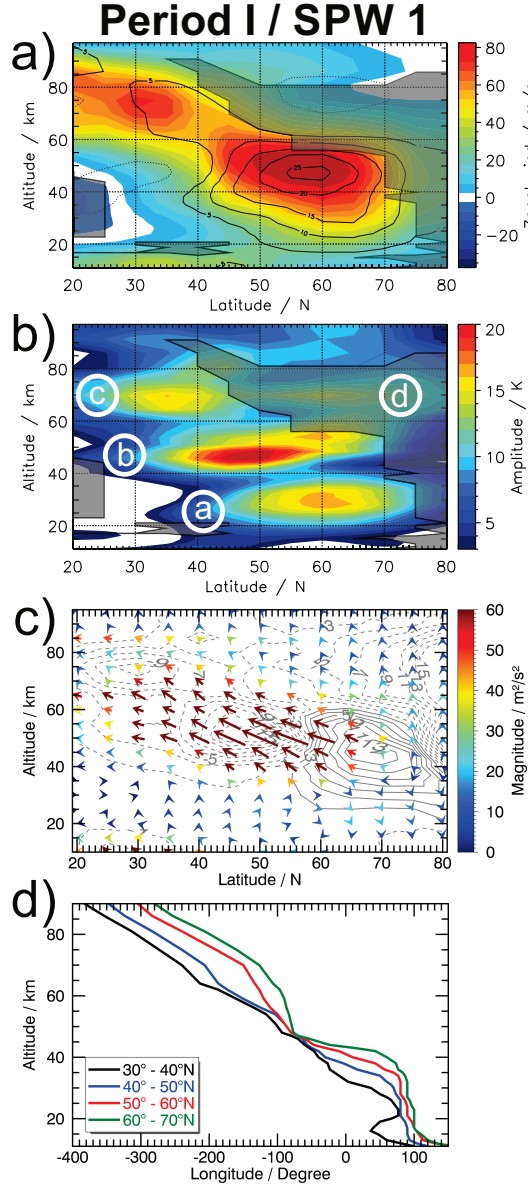

**Figure 2.** Latitude-altitude cross-sections of **a)** the zonal mean zonal wind in colored contours, the deviation of each time period from its 12-year mean in black contour lines with steps of 5 ms$^{-1}$ and the areas where the refractive index squared $n^2$ is negative are grey shaded; **b)** the amplitude of the SPW 1, areas where the refractive index squared $n^2$ is negative are grey shaded; **c)** Eliassen-Palm flux vectors of the SPW 1 with the magnitude of the flux is given by the length of the vector as well as by their color, contour lines present the EP-flux divergence in ms$^{-1}$ per day; and **d)** phase information of each SPW depending on the altitude and for different latitude bands. All data presented here are derived from MLS data.





to the negative refractive index squared (grey shaded area in Fig. 2a and 2b). So there is a PWG from the mid-latitude lower stratosphere up into the subtropical upper mesosphere.

To solve the question whether wave (c) in Fig. 2b is the same as wave (a) which propagates from the stratosphere over the stratopause region (wave (b)) into the subtropical mesosphere, we use the EPF vector. Figure 2c shows the EPF vectors of the
SPW 1 for Period I. The magnitude of the vectors is given by the color coding as well as by the length of the vectors. The EPFD is given by grey contour lines in the background. While the vectors in the mid-latitude ($40° − 60°$N) lower and middle stratosphere show dominantly upward, they are downward pointed in polar latitudes ($60° − 80°$N). In lower latitudes ($20° − 40°$N) the vectors are predominately equatorward pointed in the lower stratosphere and upward pointed above in the middle stratosphere. In the upper stratosphere and mesosphere the vectors show equatorward and upward in almost equal parts. This
results in an increase of EPFD especially in the subtropical upper mesosphere where the magnitude is three times larger than in the 12-year mean (not shown). In the upper mesosphere northward of $50°$N the EPF vectors predominantly point upward. In other words the SPW 1 generated in the lower stratosphere could be propagated upward in midlatitudes until the upper stratosphere. Then the equatorward component increases and the wave propagates upward and equatorward from the mid-latitude upper stratosphere into the subtropical upper mesosphere. So it seems that the origin of wave (c) in Period I is in the
lower stratosphere and that the conditions for vertical propagation were favorable in this period for the SPW 1.

It is known that an upward propagating SPW has a westward phase shift with height (e.g., Smith, 1997). The term phase jump is here used for very sharp changes in the longitudinal shift with height after which a vertical propagation of a SPW is not likely. The term phase kink is used for relatively smooth changes in the longitudinal shift with height after which a vertical propagation of a SPW is still likely and which is probably caused by changed propagation conditions. To finally prove that
wave (a) and (c) are the same, figure 2d shows the phase location of the SPW 1 for different latitude bands. Confirming our assumption, there is a continuous westward phase shift in vertical in the latitude band $40° − 50°$N ranging from the lower stratosphere into the upper mesosphere. A similar characteristic is found in the southernmost latitude band $30° − 40°$N above 22 km. In this latitude band (black line) there are two phase jumps below 22 km, each changing the direction of the longitudinal shift with height, caused by the subtropical jet and the area of negative refractive index right above (see Fig. 2a). Above 40 km
the two southernmost latitude bands have an almost identical and steady increase with height. Thus the SPW 1 (wave (c) in Fig. 2b) is generated in the mid-latitude lower stratosphere and propagated upward all the way into the subtropical mesosphere.

The two northernmost latitude bands show a different behavior in the vertical propagation. In particular the latitude band $60° − 70°$N has a phase jump at 50 km. Above this altitude the westward propagation with height is almost stopped for around 15 km and slowly starts again above. This feature, together with the only upward directed EPF vectors and the negative
refractive index in that area, is a strong evidence that wave (d) did not propagate from below into the upper polar mesosphere but was in situ generated by longitudinally variable dissipating GWs. Since we focus on the subtropical mesospheric SPW here, we will discuss the origin of the polar mesospheric wave (d) in more detail later in section 6.

Summarizing, the subtropical mesospheric SPW 1 (wave (c)) dominating in Period I propagates from the mid-latitude lower stratosphere into the subtropical upper mesosphere.



## 4    The origin of the subtropical mesospheric SPW 2 in Period II

The PNJ in Period II is weaker than in Period I as well as the area of strengthened zonal wind ranging from the polar stratosphere into the subtropical mesosphere (see Fig. 3a). Similar to Period I, the PNJ is up to 25 ms$^{-1}$ stronger than in the 12-year mean. However, the area is narrower and slightly tilted compared to that in Period I. There is also an increased zonal wind in the

subtropical mesosphere from 50 to 70 km that is up to 10 ms$^{-1}$ stronger than the multi year average. This region, however, is somewhat more separated from the PNJ than in the case during Period I.

The amplitude of the SPW 2 shifts only slightly southward with height below 50 km and much stronger above (see Fig. 3b). The area where no SPW propagation can occur (grey shaded area) is similar to that of SPW 1 during Period I northward of 50°N, but the tail into the subtropical upper mesosphere is replaced by a small area around 80 km between 40° – 45°N. The

southward shift of the SPW 2 is again in accordance with the area in which the SPW 2 can not propagate. So there is a PWG from the mid-latitude lower stratosphere into the subtropical and mid-latitudinal upper mesosphere (the PWG is bounded by the grey shaded area in Fig. 3a and 3b).

Figure 3c shows the EPF vectors of the SPW 2 for Period II. The vectors in the mid-latitude (50° – 65°N) lower and middle stratosphere show predominantly upward, while they are upward and poleward pointed in polar latitudes (65° – 80°N)

and predominantly equatorward pointed in lower latitudes (20° – 45°N) in the lower and middle stratosphere. In the upper stratosphere and mesosphere the vectors show equatorward and upward in almost equal parts southward of 60°N. This results in an increase of EPFD especially in the subtropical mesosphere where the magnitude is three times larger than in the 12-year mean (not shown). Note that the magnitude of the EPF vectors and of the EPFD is smaller compared to that of the SPW 1 in Period I. In the upper mesosphere northward of 60°N the EPF vectors predominantly point upward. In other words the SPW 2,

likely generated in the lower stratosphere, is able to propagate upward in mid-latitudes until the middle stratosphere. Then the equatorward component increases and the wave propagates upward and equatorward from the mid-latitude middle stratosphere into the subtropical mesosphere.

The westward phase tilt with height of the SPW 2 in Period II is not as strong as it is for the SPW 1 in Period I (see Fig. 3d). Below 35 km, there is no westward shift in vertical for the two northernmost latitude bands (50° – 70°N) meaning that the

SPW 2 might be barotropic, and there is only a light westward phase tilt with height for the two southernmost latitude bands (30° – 50°N). Again, in the lower stratosphere the latitude band 30° – 40°N shows a phase jump with eastward shift in vertical caused by the subtropical jet. Above 40 km the magnitude of the westward phase tilt with height increases almost steadily for the two southernmost latitude bands, but there is a phase kink at 60 km again. Above this kink the two southernmost latitude bands split and the 30° – 40°N latitude band increases the westward shift with height again. The 50° – 60°N latitude band has

a phase kink at 50 km, followed by almost no longitudinal shift in vertical for 20 km. The 60° – 70°N latitude band even has a phase jump at 50 km resulting in an eastward shift with height for the overlying 20 km. Thus we assume that the northern part of wave (d) is in situ generated by longitudinally variable breaking GWs, which is supported by the positive EPFD in the polar mesosphere. Since we focus on the subtropical mesospheric SPW here, we will discuss the origin of wave (d) later in section 6 in more detail.



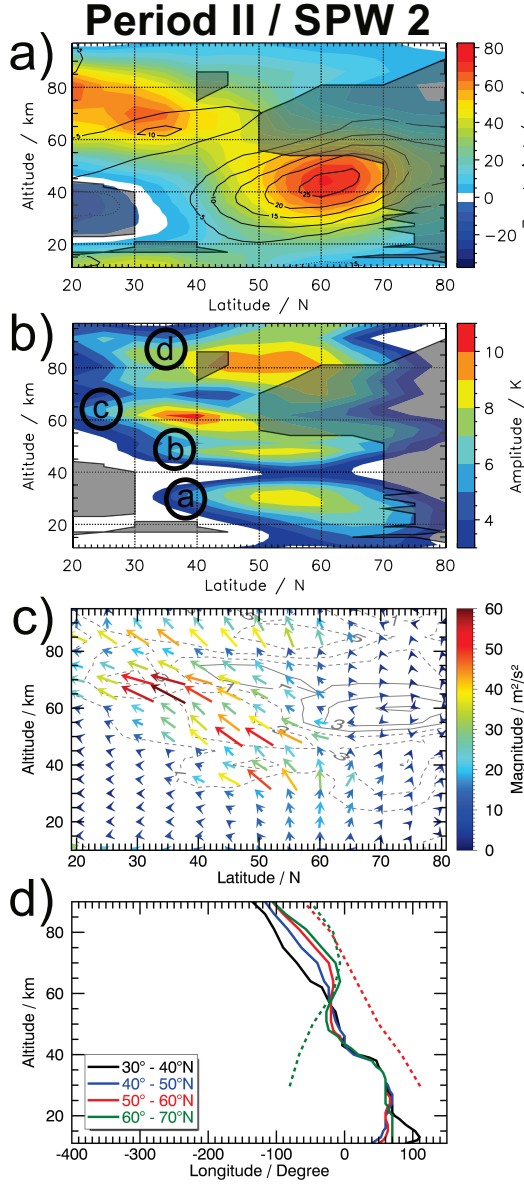

**Figure 3.** Same as Fig. 2 but for the SPW 2 in Period II. In addition, Fig. 2d shows the phase information of the GW drag filtered for SPW 2 depending on the altitude and for different latitude bands (dashed curves). The GW drag data are derived from SABER observations.

Taking together the information from the EPF vectors and from the phase shift with height, it was shown that wave (c) is generated in the mid-latitude lower stratosphere and propagated upward all the way into the subtropical and mid-latitude mesosphere in Period II.





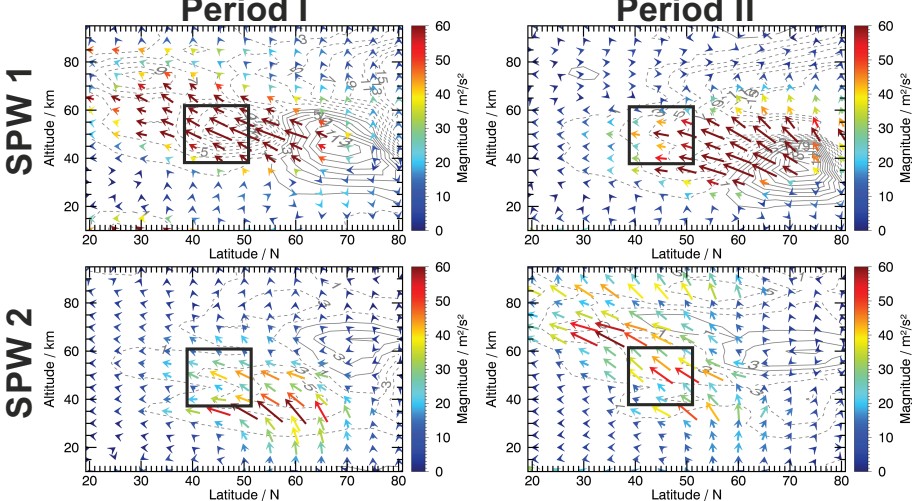

**Figure 4.** Same as Fig. 2c and 3c but also for the SPW 2 in Period I and SPW 1 in Period II. The black rectangles mark the areas crucial for upward propagation of the SPWs into the subtropical mesosphere.

Summarizing, in both time periods the respective SPW was able to propagate from the mid-latitude lower stratosphere equatorward and upward into the subtropical upper mesosphere, guided by the strong PNJ. Surprisingly, in Period II the dominating wavenumber of the SPW changes from 1 to 2 in the subtropical mesosphere, although the conditions for upward propagation were favorable for the SPW 1 too. This change in the dominating wavenumber is studied more closely in the next

section.

## 5    Why dominates the SPW 2 in Period II?

To investigate the reason for the change in wavenumber of the subtropical SPW in the mesosphere from Period I to II, Fig. 4 shows the EPF vectors and divergence of the SPW 1 (top) and SPW 2 (bottom) for Period I (left) and Period II (right). The magnitude of the EPF vectors of the SPW 1 in the subtropical mesosphere is much larger in Period I than in Period II as

expected. The accumulation of large EPF vectors is shifted poleward in Period II compared to Period I and especially in mid latitudes the upward component is almost completely missing in Period II.

The behavior of the SPW 2 in the two considered time periods is reversed to that of the SPW 1. The magnitude of the EPF vectors of the SPW 2 in the subtropical mesosphere is much larger in Period II than in Period I, and the accumulation of enhanced EPF vectors is shifted southward in Period II compared to Period I. Similar to the SPW 1 in Period II, the upward

component of the EPF vectors for the SPW 2 is missing in mid latitudes in Period I. Combining these results, it seems that the area from 40° – 50°N and from 40 to 60 km (black rectangles in Fig. 4) is crucial for the upward propagation of the SPWs into the subtropical mesosphere.





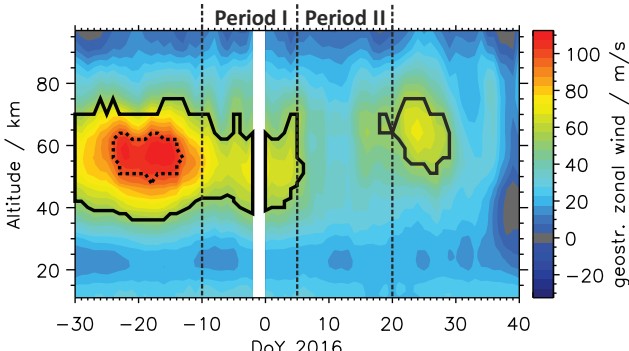

**Figure 5.** Time-altitude cross-sections of the zonal mean zonal wind averaged between $40°$ and $50°$N. Areas enclosed by dotted (solid) black lines present areas with a negative refractive index squared $n^2$ for the SPW 1 (2), i.e. areas where the vertical propagation of the SPW is prohibited. Data are derived from MLS.

Figure 5 shows the temporal evolution of the zonal mean zonal wind averaged between $40° - 50°$N. The areas enclosed by the black dotted (solid) lines are characterized by a negative refractive index squared $n^2$ for the SPW 1 (2). Focusing on the altitude range 40 to 70 km, the zonal mean zonal wind is very strong with values of up to 100 ms$^{-1}$ in early and mid December. During this period the conditions are not favorable for upward propagation of the SPW 1 and 2. Afterwards a
two step weakening occurs starting shortly before Period I. In the first step the zonal wind decreases down to approximately 70 ms$^{-1}$ allowing the SPW 1 to propagate upward, but not SPW 2 since the refractive index squared is still negative for the SPW 2. In the second step the wind further decreases, now down to 30 ms$^{-1}$, allowing the SPW 1 and 2 to propagate upward. Thus the SPW 2 was not able to propagate upward during Period I due to the strong wind at $40° - 50°$N between 40 and 60 km. The zonal wind weakening from Period I to Period II was caused by the SPW 1 in Period I. Looking again at Fig. 4 there is
a strong EPF convergence of the SPW 1 in Period I decelerating the westerly wind especially in the area between $40° - 50°$N and between 40 and 60 km. This zonal wind deceleration paves the path for the upward propagation of the SPW 2 in Period II since the upward propagation of the SPW 2 is only able in a weak zonal mean zonal wind. It is now clear why the SPW 1 dominates in Period I in the subtropical mesosphere, but it is still not clear why the SPW 2 dominates in Period II because it was possible for the SPW 1 to propagate upward in Period II, too.
Figure 6 shows the temporal evolution of the vertical component of the EPF vector $F_z$ for the SPW 1 averaged between $40° - 50°$N (top) and $60° - 70°$N (bottom). In Period I there is a strong enhancement in $F_z$ at $40° - 50°$N which vanishes in Period II. As expected, $F_z$ is weak in Period I at $60° - 70°$N but, surprisingly, it increases significantly in Period II. This means that in Period II the SPW 1 was able to propagate upward in polar latitudes probably due to the weakened PNJ (cf. Fig. 2a and 3a). So it seems that now when the SPW 1 can propagate upward in polar latitudes there is no need anymore to do so at lower latitudes.
This stronger upward propagation of the SPWs in the northern channel of the PW guide compared to the southern channel is in accordance with the climatology (e.g., Albers et al., 2013).




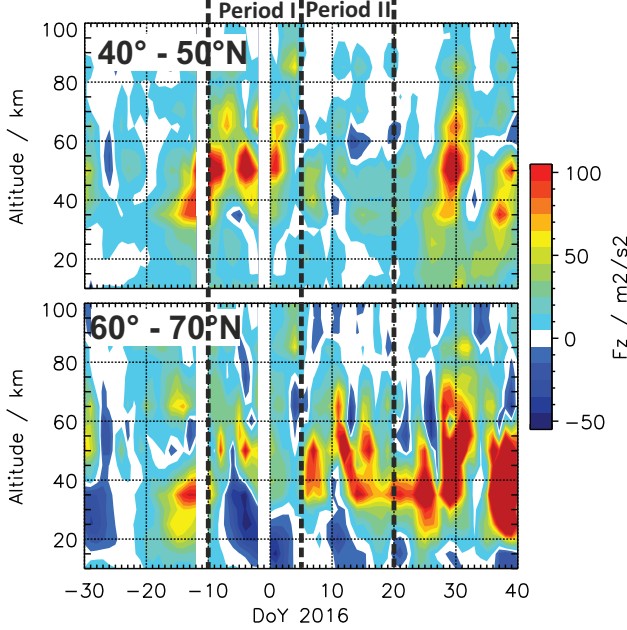

**Figure 6.** Time-altitude cross-sections of the vertical component of the EP-flux vector $F_z$ for the SPW 1 averaged between $40° - 50°$N (top) and $60° - 70°$N (bottom). Data are derived from MLS.

Another possible cause for the SPW 2 strengthening in Period II is the disruption of the QBO (Osprey et al., 2016). Due to this disruption, the QBO is in its westerly phase again/still. From Holton and Tan (1980) we know that from January to March the SPW 2 is stronger during the westerly phase of the QBO. Thus it could also be that the disruption of the QBO has an impact on the dominating SPW in the subtropical mesosphere.

Summarizing, the SPW 1 dominates in Period I in the subtropical mesosphere since the SPW 2 was not able to propagate upward due to the strong zonal wind at $40° - 50°$N in the upper stratosphere and lower mesosphere. Both SPWs were theoretically able to propagate upward into the subtropical mesosphere in Period II since the convergence of the SPW 1 in Period I weakens the zonal mean zonal wind and therefore paves the path for the upward propagation of the SPW 2 in Period II. Another cause for the change in the dominating wavenumber between Period I and II might be the disruption of the QBO, remaining in the westerly phase in January 2016 resulting in a stronger SPW 2 during this QBO phase. Due to the weaker PNJ in Period II the SPW 1 was now able to propagate upward into the mesosphere at polar latitude which, as we assume, makes it not necessary anymore to do this at lower latitudes.

As mentioned before, the winter 2015/16 was also characterized by an outstanding strong El Niño event (Palmeiro et al., 2017, and references therein). In principle there are two types of El Niño events which have different impact on the polar stratosphere. While El Niño events confined to the east Pacific ocean result in a warm and weak polar vortex (Garcia-Herrera et al., 2006), an El Niño event confined to the central Pacific ocean leads to a colder polar stratosphere and a stronger polar



vortex (Iza and Calvo, 2015). Previous strong El Niño events were confined to the east Pacific ocean but that one in winter 2015/16 was extended to the central Pacific and west of date line (e.g., Parker et al., 2016). Thus the extraordinarily strong polar vortex in early winter 2015/16 as reported by Matthias et al. (2016) might be caused by the untypical structure of the strong El Niño in that winter. Hence, this strong polar vortex affected by the strong El Niño event helped to guide the SPWs
into the subtropical mesosphere.

   The QBO is usually driven by a combination of drag exerted by global scale equatorial wave modes and by tropical GWs (Baldwin and Dunkerton, 2001; Ern et al., 2014, and references therein). The unusual disruption of the QBO in winter 2015/16, characterized by anomalous easterly acceleration occurring in the QBO westerlies (e.g., Osprey et al., 2016; Newman et al.,
2016; Coy et al., 2017), starts approximately at the same time as the unusual strong SPW amplitude occurs in the subtropical mesosphere. Osprey et al. (2016) found, as a primary source for the QBO disruption, an anomalous easterly acceleration of the equatorial winds at 40 hPa (approx. 23 km) caused by waves propagated to the equator from the northern hemisphere in the region below. For the SPW 1 we found an enhanced equatorward EPF at altitudes below 25 km and at latitudes $20° - 40°$N during Period I together with an EPF convergence (cf. Fig. 4) resulting in an easterly acceleration of the zonal wind. In Period
II, the magnitude of this equatorward EPF is lower compared to Period I and occurs at altitudes below 15 km in the EPF of the SPW 1 and SPW 2. So it seems that there is another PWG at lower latitudes vertically capped by the negative refractive index induced by the subtropical jet. This PWG guides the PWs into the equatorial region below 20 km. These findings confirm the model results of Osprey et al. (2016) but need further investigations.
   However, Coy et al. (2017) found also large tropical momentum flux divergences in other years without a reversal of the
QBO. Comparing the zonal mean zonal wind behavior in the subtropics between Period I and II (cf. Fig. 2a and Fig. 3a) shows that the westerly wind field in the mesosphere moved down by 5 km and that the stratospheric easterly wind field shrinks about 5 km in the vertical. This downward movement and shrinking caused by the breaking SPWs in the subtropical mesosphere might have had an impact on the development on the QBO disruption. Thus, it is not clear whether the QBO disruption causes or is caused by the unusual SPW amplitude in the subtropical mesosphere. However, a detailed study and discussion on this
hypothesis is beyond the scope of this paper.

## 6   Where is the origin of the polar mesospheric SPWs?

At first we want to focus on the origin of the SPW 1 wave (d) in Period I located between $50° - 70°$N at 65 to 75 km with its maximum at $60°$N at 70 km (see Fig. 2b). This wave is situated completely in an area of negative refractive index (see Fig. 2b) which means that this wave can not propagate and that it is more a stationary wave-like structure caused by in situ processes
rather than a physical SPW 1 which is vertically propagating. There are two possibilities for the development of this wave-like structure. One is the already mentioned in situ generation by longitudinally variable GW drag (Smith, 2003; Lieberman et al., 2013). The other one was postulated but not proven by Siskind et al. (2010) who investigated an increased wave 1 amplitude





above a region with easterly wind and negative meridional gradient of the potential vorticity in the mesosphere. This negative meridional gradient of potential vorticity indicates potential instability which might generate a wave 1 structure.

Since SABER is in the southern yaw cycle until early January we are not able to prove that wave (d) is generated by longitudinally variable GW drag in Period I. However, the longitude-altitude cross-section of the wavenumber 1 filtered zonal
wind shows an anti correlation between the stratospheric and mesospheric zonal wind disturbance (not shown) indicative for a SPW propagation from below or an in situ generation by GWs (Smith, 2003). Since we already ruled out the SPW propagation from below due to the negative refractive index in that area and the downward pointing EPF vectors below, the in situ generation by longitudinally variable GW drag remains after the theory of Smith (2003). This assumption is also supported by the phase jump at 50 km in the latitude band $60° - 70°$N with almost no longitudinal propagation with height for 10 km (see Fig. 2d)
and a non-uniform GW drag at $50°$N with a dominating wave 1 structure (not shown).

However, in the same area where wave (d) occurs in Period I the meridional gradient of the potential vorticity is negative (not shown) which is a necessary condition for barotropic and/or baroclinic instabilities and might be indicative for an in situ generation of SPWs by instabilities (Siskind et al., 2010). Iida et al. (2014) showed instabilities forming SPWs might be brought about by an intensification of the PNJ in the stratosphere and the subtropical westerly jet in the mesosphere which is
the case in Period I (cf. Fig. 2a). Thus it is also possible that wave (d) in Period I is (partially) in situ generated by instabilities induced by the strong westerly jets in the mid latitude stratosphere and subtropical mesosphere.

Since we can not prove whether wave (d) is in situ generated by longitudinally variable GW drag or by instabilities further investigations are needed but they are beyond the scope of this paper.

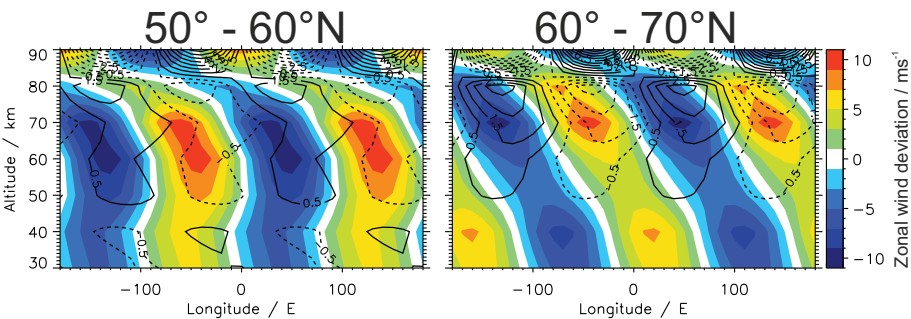

**Figure 7.** Latitude-altitude cross-sections of the geostrophic zonal wind amplitude (colored contour) and the absolute GW drag amplitude in m/s/d (contour lines). The plot shows the deviation from the zonal mean of the wavenumber 2 filtered atmospheric wind component and GW drag. Data are derived from SABER.

The SPW 2 wave (d) in Period II ranges from $30°$ to $70°$N between 70 and 90 km with its maximum between $40° - 60°$N around 80 km (see Fig. 3b). The northern part of this wave lies above an area of negative refractive index $n^2$ (see Fig. 3c). There is another small area with a negative refractive index at $40° - 45°$N around 80 km. Between the large and small areas of negative refractive index and southward of the small area there are only small positive values of the refractive index




(not shown). Additionally, the EPF vectors at 80 km around the maximum of wave (d) are very small in the whole northern hemisphere. Thus, wave (d) could not propagate from below into the mesosphere.

The northernmost part of the SPW 2 wave (d) is very likely in situ generated by longitudinally variable GW drag. Therefore we investigate the longitudinal structure of absolute GW drag that is obtained from vertical gradients of absolute GW momentum fluxes that are derived from SABER temperature observations (see also Ern et al., 2011, 2016, and references therein). This parameter does not provide directional information unless atmospheric background conditions allow for assumptions to be made and it can be used as a proxy for "real" GW drag. Fig. 7 shows the zonal wind and SABER GW drag each filtered for wavenumber 2 for the two northernmost latitude bands. Note that the filtering of the wavenumber 2 is reasonable for these two latitude bands, since the zonal wind is westerly at all longitudes below 60 km (not shown). The GW spectrum should therefore be dominated by waves of westward directed phase speeds (opposite to the background wind). For this reason, as a working hypothesis, we assume that the GW drag is negative everywhere (similar as in the simulations by Holton, 1984), and we will show that this leads to an overall consistent picture. In the case of the two southernmost latitudes bands, the zonal wind is dominating westerly with some small areas of easterly wind (not shown). Thus, we cannot be sure about the sign of the GW drag in the mesosphere which makes it difficult to filter for wavenumber 2.

In the 60°N – 70°N latitude band the maximum of the GW drag lies within the mesospheric minimum of zonal wind. In addition, the longitudinal phase tilt with altitude of the filtered GW drag is similar to the phase tilt of the SPW 2 (see Fig. 3d). Since the GW drag decelerates the zonal wind and the SPW 2 propagation is not possible, the wave 2 structure in the polar mesosphere primary originates from the longitudinally variable GW drag which is consistent with studies from Smith (2003) and Lieberman et al. (2013). Even though we can not prove the in situ generation by longitudinally variable GW drag, all necessary preconditions are given: 1) the SPW can not propagate from below, 2) a non-uniform distribution of the GW drag is observed at the altitude where the SPW generation takes place, and 3) the forcing takes place in regions of weak background wind, which is in agreement with Smith (2003).

In the other latitude band (50°N – 60°N) the minimum of the zonal wind is westward shifted compared to the GW drag maximum at 70 to 80 km and the longitudinal phase tilt of the GW drag coincides with the SPW 2 phase tilt above 70 km. Thus in this latitude band the in situ generation is not only caused by longitudinally variable GW drag. We assume that the impact of the longitudinally variable GW drag decreases with decreasing latitude based on increasing uniformity with decreasing latitudes (not shown) and on the decreasing wavenumber 2 amplitude of the absolute GW drag with decreasing latitude in the altitude range 70 – 80 km (see Fig. 7). Another possible in situ generation mechanism is the barotropic and/or baroclinic instability. In Period II the meridional potential vorticity gradient is negative where the refractive index is negative, roughly speaking (not shown). This indicates an in situ generation of SPWs by instabilities (Siskind et al., 2010). Furthermore, similar to Iida et al. (2014) the stratospheric PNJ and the mesospheric subtropical jet were intensified possibly bringing about the instabilities in the mesosphere and hence the possibility to generate SPWs in the mesosphere.

The mix in the origin of mesospheric SPW is in agreement with model studies of Smith (2003) and observational studies of Lieberman et al. (2013) whereby these studies focused on the upward propagation and in situ generation by longitudinally





variable GW drag only. Our study indicates that a mix of in situ generated SPW by longitudinally variable GW drag and barotropic and/or baroclinic instabilities is also possible.

## 7 Summary

This paper investigated the origin of mesospheric SPWs in a case study during one month in mid winter 2015/16 where an unusually strong SPW 1 and 2 amplitude was observed in the subtropical mesosphere. While during the first half of the period (late December 2015 to early January 2016) the SPW 1 dominates in the subtropical mesosphere, it is the SPW 2 in the second half (early January to mid January 2016). At the same time there is also an increased SPW amplitude of the respective SPW in both periods in the midlatitude and polar mesosphere.

The origin of the subtropical mesospheric SPWs is located in each period in the mid latitudinal stratosphere. We find that the SPW 1 in Period I as well as the SPW 2 in Period II propagated upward and equatorward from the mid latitudinal stratosphere into the subtropical mesosphere, guided by the unusual strong PNJ (e.g., Matthias et al., 2016). While the strong PNJ might be influenced by the strong El Niño, the SPWs might be influenced by or had an impact on the development of the disruption of the QBO starting in the same time period.

The change in the dominating wavenumber from Period I to Period II arises the question why this change occurred at all. We localized the area from 40°N to 50°N and between 40 and 60 km especially being crucial for upward propagation from the mid latitude stratosphere into the subtropical mesosphere. In this area the upward propagation of the SPW 2 was prohibited in Period I due to too strong westerly winds in that region. These strong westerly winds were decelerated in Period I by the SPW 1 interacting with the mean flow, thereby paving the path for upward propagation of the SPW 2 in Period II. However, the upward propagation of the SPW 1 in Period II was much weaker in this crucial area compared to Period I although an upward propagation was theoretically possible. This can be explained by a poleward shift of the SPW 1 activity induced by the weakened zonal mean PNJ and thus strengthened polar channel of the PW guide. It is also possible that the increase in the SPW 2 amplitude in Period II is influenced by the disrupted QBO resulting in a recurring westerly phase which increases the SPW 2 activity in general (Holton and Tan, 1980).

The polar mesospheric SPWs are in situ generated likely by a mixture of longitudinally variable GW drag and barotropic and/or baroclinic instabilities. This mixture is dominated by longitudinally variable GW drag towards polar latitudes and instabilities towards lower latitudes resulting in a smooth transition from pure in situ generation by longitudinally variable GW drag in polar latitudes and pure in situ generation by instabilities in the subtropical upper mesosphere. With this observational study we showed that the origin of mesospheric SPWs can be the upward propagation of SPWs as well as the in situ generation by longitudinally variable GW drag or instabilities and that all three mechanisms can occur at the same time which is partially in accordance with the model study of Smith (2003).





*Acknowledgements.* The work by Manfred Ern was partly supported by the Deutsche Forschungsgmeinschaft (DFG, German Research Foundation) project ER 474/4-2 (MS-GWaves/SV) which is part of the DFG researchers group FOR 1898 (MS-GWaves). We thank the Jet Propulsion Laboratory/NASA for providing access to the Aura/MLS level 2 retrieval products downloaded from http://mirador.gsfc.nasa.gov. SABER data were provided by GATS Inc. and are freely available at http://saber.gats-inc.com. The authors would like to thank the teams of the MLS and SABER instruments for their effort in providing and continuously improving the high-quality data sets used in this study. A big thanks goes to Axel Gabriel and Lena Schoon for their very helpful discussions and comments. We also want to thank Dieter H.W. Peters for helpful comments.



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
