# Peer review of "On the origin of the mesospheric quasi-stationary planetary waves in the unusual Arctic winter 2015/16"

_Atmospheric Chemistry and Physics, 2017_

## Referee Comment (RC1) · Anonymous Referee #1 · 7 Jan 2018

Comments on "On the origin of the mesospheric quasi-stationary planetary waves in the unusual Arctic winter 2015/16" by Vivien Matthias and Manfred Ern

This paper presents a detailed investigation of the spatial distribution and the propagation features of the SPW1 and SPW2 observed during the boreal mid winter of 2015/2016 that was characterized by an unusually strong polar night jet (PNJ). The both satellite temperature MLS/Aura and SABER/TIMED data for the period of time between 21 Dec 2015 and 20 Jan 2016 have been used for studying the characteristics of the SPWs and the GW drag respectively. The authors found extraordinary large SPWs in the subtropical mesosphere as the SPW1 dominates from late December

2015 to early January 2016 (named as Period I) while the SPW2 dominates from early January until mid January 2016 (called as Period II). Moreover, the authors found also the amplification of the SPW1 in the Period I and the SPW2 in the Period 2 respectively in the polar mesosphere. These results actually defined the basic object of the study that is to define the origin of the observed mesospheric subtropical and polar SPWs. By using all possible data analysis methods: (i) two-dimensional least squares method for determining the SPW amplitudes and phases; (ii) calculation of the geostrophic winds for estimation of the propagation conditions of the SPWs; (iii) diagnostic analysis (calculations of: the refractive index squared $n^2$; the Eliassen-Palm flux (EPF) vectors and their divergence, and the meridional potential vorticity gradient) for distinguishing regions of wave propagation from wave evanescence, defining the direction and strength of SPW propagation, the interaction of the SPWs with the mean flow and the condition for barotropic and/or baroclinic instabilities respectively; (iv) longitude distribution of the GW drag. By the synergy of the all above mentioned analysis methods and satellite data sets, the authors found clear evidences for the origin of the both subtropical SPWs (they propagate from the mid-latitude stratosphere) and polar SPWs (generated in situ by longitudinally variable GW drag and by instabilities) in the mesosphere and in the two periods. Additionally, the authors considered also a possible contribution of the unusually strong SPWs in the subtropical mesosphere to the disruption of the QBO in the same winter as well as the impact of the strong El Niño on the enhancement of the PNJ.

I find this paper very interesting and useful particularly for colleagues working on the PW coupling processes. The clarification of the area that is crucial for the upward propagation of the SPWs into the subtropical mesosphere is an important new result. The topic of the paper is certainly appropriate for the journal. It is written very clearly and presents informative figures of high quality. The abstract adequately presents the obtained in the paper results. Therefore I suggest the publication of this paper after addressing only two very minor comments mentioned below.

(i) P. 7: It is written: "…. as wave (a) which propagates from the stratosphere over the stratopause region (wave (b))……", or "In other words the SPW 1 generated in the lower stratosphere could be propagated upward in midlatitudes until the upper stratosphere." Similar statements have for the SPW2 as well in p. 8. Yes, the SPW phase analysis and EPF vectors indicate vertical and equatorward propagations however the presence of the waves (a) and (b) from Figures 2 and 3 represents actually the typical double-peak altitude structure of the SPWs in the field of the temperature. This issue is reported by Pancheva et al. (2009; please, see Figure 9 there) and is a consequence of the hydrostatic equation (Sassi et al., 2002). Moreover, this double-peak altitude structure is valid not only for the SPWs but for all PWs in the field of the temperature; for example, Pancheva et al. (2016) showed this feature for the quasi-2-day waves.

(ii) I have some doubt about wave (d) from Fig. 3 that it is in situ generated. I think that the waves (c) and (d) represent the above mentioned double-peak altitude structure of the SPW2 in the field of the temperature. This could be checked by considering the SPW2 but in the field of the geopotential height; the latter should have a single peak maximum situated at an altitude coinciding approximately with the altitude of the minimum between the double-peak structures in the temperature. Both the phase structure and EPF vectors southward from 60°N show vertical and equatorward propagation of the SPW2; wave (d) is also above the region where n2 is negative.. I agree that EPF vectors are not large at altitude of 80 km but below and above this altitude they are quite large. I think also that the barotropic and/or baroclinic as well as the GW drag may additionally strengthen the northern part of wave (d).

Typos: The text of Figure 7: Latitude-time…. should be Longitude-time….

REFERENCES: D. Pancheva, P. Mukhtarov, B. Andonov, N.J. Mitchell, J.M. Forbes (2009), Planetary waves observed by TIMED/SABER in coupling the stratosphere–mesosphere–lower thermosphere during the winter of 2003/2004: Part 2—Altitude and latitude planetary wave structure, 71, 75-87, doi:10.1016/j.jastp.2008.09.027.

Sassi, F., R. R. Garcia, B. A. Boville, and H. Liu (2002), On temperature inversions and the mesospheric surf zone, J. Geophys. Res., 107(D19), 4380, doi:10.1029/2001JD001525.

Pancheva, D., P. Mukhtarov, D. E. Siskind, and A. K. Smith (2016), Global distribution and variability of quasi 2 day waves based on the NOGAPS-ALPHA reanalysis model, J. Geophys. Res. Space Physics, 121, doi:10.1002/2016JA023381.
* * *

---

## Referee Comment (RC2) · Anonymous Referee #2 · 23 Jan 2018

This paper on the origin of the mesospheric planetary waves in the Arctic winter of 2015/2016 is well written and organized. The scientific reasoning is sound and methods are appropriate. The only issue I have is that I am not convinced that the authors showed sufficient evidence that wavenumber 2 is generated by in situ gravity wave drag at the high latitudes. If the authors address this I think it should be published with the following minor revisions.

Page 1, line 15: change to "...show that all three mechanisms..."

Page 1, line 24: change to "In addition to these global anomalies..."

Page 3, line 8: change to "longitudinally variably" or "variably in longitude"

[Figure]

Page 3, line 10: change the word order to "...how favorable the conditions are in the..."

Page 3, line 11: awkward word order, change to "... in situ generation of the quasi 2-day wave, for example."

Page 4, line 3-4: change "in all latitudes..." to "at all latitudes"

Page 4, line 14: should be "atmospheric parameters"?

Page 6, Figure 2: Yes the PNJ is much stronger than the 12-year mean, but is this surprising? Doesn't the NH PNJ move around a lot from year to year, so that the 12-year mean is of course a little washed out? It would make a stronger case that this PNJ is exceptional if the figure showed the spread of all 12 years instead of just the mean. So for example, a line plot of the zonal mean zonal wind averaged between 50 and 60 N and 40 and 60 km versus time for the entire Period I.

Page 7, line 17: change word order to "used here"

Page 7, line 21: change "shift in vertical" to "shift in the vertical" or "vertical shift"

Page 8, line 24 and 26: change to "in the vertical"

Page 10, section 5 heading: change to "Why does the SPW 2 dominate in Period II?"

Page 11, line 12: change to "...only possible in a weak zonal mean zonal wind."

Page 14, Figure 7 caption: The date range of the SABER data should be added to the caption. Is it the entire Period II? This was not clear in the text either.

Page 14, Figure 7: I'm not convinced that it makes sense to filter the GW drag for wavenumber 2. If wavenumber 2 is the dominant wavenumber in the zonal wind, and if the GW drag is the cause of this, then the unfiltered GWD should show that. It might be more interesting to see what the unfiltered GWD looks like, because the wind doesn't just feel the wavenumber 2 GWD. The numbers are also very small for the GWD. On the order of 0.5 to maybe 2 m/s/d it looks like, an order of magnitude smaller than those

from Smith 2003 for example. Again, it might be more fair to show the GWD from all wavenumbers.

Page 15, line 18: change "primary" to "primarily"

Page 16, line 14: change to "raises the question"

---

## Referee Comment (RC3) · Anonymous Referee #3 · 29 Jan 2018

This is an interesting paper reporting an investigation of large amplitude stationary planetary waves observed in the mesosphere during the winter of 2015/16. This was a time when there was an unusually strong polar night jet. The origins of the stationary planetary waves of wavenumber 1 and 2 (i.e., SPW1 and SPW2) are investigated in detail in the context of the excitation and propagation of the waves. It is suggested that the high-latitude SPW2 is forced by longitudinally-variable gravity-wave drag, but that at lower latitudes, this forcing is increasingly replaced by barotropic/baroclinic instabilities, which eventually dominate by subtropical latitudes. The authors make use of absolute gravity-wave momentum fluxes derived from SABER observations. The paper is a nice example of the powerful insight into gravity-wave forcing of the atmosphere that can be

provided by such measurements.

The paper is very well written, clearly argued and presents scientific results of considerable interest and of the highest quality. Overall, this is an excellent study which fully deserves publication in ACP.

I have two particular requests for changes before the paper can be accepted for publication.

My first request is that the authors expand on their explanations of the SABER measurements. In particular, there should be a more expansive description of the technique and its limitations. At present the measurements are simply described as being of "absolute GW momentum fluxes" and the reader is provided with references for more substantive explanation of the technique.

However, the SABER measurements are a central part of the paper and there should be i) a paragraph of explanation describing the technique at the point where it is introduced and ii) some discussion of the limitations of the technique. In respect of the latter, I believe that these GW momentum flux measurements actually yield a lower bound rather than a fully-constrained value, since the estimates of horizontal wavelength depend on the angle between the satellite's orbit and the phase fronts of particular gravity waves.

The arguments on p14 in paragraph 2 about the longitude-altitude cross section of wavenumber 1 filtered winds and the non-uniform GW drag at 50N with a wavenumber 1 structure refers to figures "not shown" – these figures would confirm the arguments being made by the authors so they should be included.

The arguments about the wavenumber 2 component of gravity-wave drag associated with Fig 7 would be strengthened by some explanation of the total drag and its other component wavenumbers. There does seem to be a wavenumber 2 component as shown, but how big is it compared to the zonal-mean value and the other wavenumbers?

My second request is that some of the figure be made larger. As presented, some of the contours in Figs 2c, 3c, 4 and 7 are very faint, hard to read and/or close together. I think that just making the figures larger would solve this problem

MINOR POINTS

P1, l15, suggest "...show that all three mechanisms..."

Figure 1 caption, suggest "...from MLS temperature..."

P3, l21, the final sentence "The vertical propagation of... (Lin, 1982)" would make much more sense if put at the start of that paragraph.

P4, l17, should be "...winds are needed".

P4, l20, should this be $a^{-1} \partial/(\partial \hat{a}\acute{L}\check{E})$ ?

P5, l5, suggest "...which results in a westerly wind..."

P5, l9, suggest "...and following it up into..."

P5, l14, suggest "...TIMED satellite and measures temperatures..."

P5, l16, suggest "...geometries about every 60 days. For the period of..."

P8, l2, suggest "...Period I, as is the areas of..."

P11, Fig 5, what causes the missing data at days -2 to 0?

P14, l3, suggest "...not able to investigate whether wave (d)..."
* * *

---

## Author Comment (AC1) · 4 Mar 2018

Dear Referee #1,

we are grateful for your friendly and constructive review. Based on your comments and suggestions the manuscript is now improved. In the following point-by-point responses the reviewer comments are in italics, our responses are in blue.

(i): P. 7: It is written: "... as wave (a) which propagates from the stratosphere over the stratopause region (wave (b)). . .. . .", or "In other words the SPW 1 generated in the lower stratosphere could be propagated upward in midlatitudes until the upper stratosphere." Similar statements have for the SPW2 as well in p. 8. Yes, the SPW phase analysis and EPF vectors indicate vertical and equatorward propagations however the presence of the waves (a) and (b) from Figures 2 and 3 represents actually the typical double-peak altitude structure of the SPWs in the field of the temperature. This issue is reported by Pancheva et al. (2009; please, see Figure 9 there) and is a consequence of the hydrostatic equation (Sassi et al., 2002). Moreover, this double-peak altitude structure is valid not only for the SPWs but for all PWs in the field of the temperature; for example, Pancheva et al. (2016) showed this feature for the quasi-2-day waves.

We thank the reviewer for this helpful note. We compared our temperature related amplitude results with that of the geopotential height (GPH) (see below). We found that the minimum between our wave (a) and (b) corresponds to the maximum of the stratospheric GPH wave in Figure 1, 2 and 3. Thus, wave/maximum (a) and (b) in the temperature amplitude belong to the same wave. We know that the calculation of the PW amplitudes from temperature is less common due to the double-peak issue. However, in our study it is useful since one can better retrace the path of the SPW into the subtropical mesosphere and the amplitude itself is better visible in the mesosphere. We added a note on the double-peak vs. single-peak relationship between temperature and GPH amplitudes on page 2 line 15-20 and line 25 and included the information that the temperature amplitude maxima (a) and (b) belong to the same wave. Additionally we added a GPH-Version of Figure 1 into the supplements (see Figure S1).

Figure 1: Latitude-Altitude cross-section of the amplitude of the SPW 1 and 2 in Period I and II. The amplitudes are calculated using GPH data from MLS.

Figure 2: Same as Figure 1 but estimated from MLS temperature data

(ii): I have some doubt about wave (d) from Fig. 3 that it is in situ generated. I think that the waves (c) and (d) represent the above mentioned double-peak altitude structure of the SPW2 in the field of the temperature. This could be checked by considering the SPW2 but in the field of the geopotential height; the latter should have a single peak

maximum situated at an altitude coinciding approximately with the altitude of the minimum between the double-peak structures in the temperature. Both the phase structure and EPF vectors southward from 60_N show vertical and equatorward propagation of the SPW2; wave (d) is also above the region where n2 is negative.. I agree that EPF vectors are not large at altitude of 80 km but below and above this altitude they are quite large. I think also that the barotropic and/or baroclinic as well as the GW drag may additionally strengthen the northern part of wave (d).

The reviewer is partly right. Wave (c) and (d) of the SPW 2 in Period II are the same southward of approximately 45°N. However, in the polar latitudes there is a different wave which very probably does not belong to wave (c) in the subtropical mesosphere. We added a note on page 2 line 27-31 regarding the south-north splitting of wave (d) and only referred to the northern part of wave (d) when talking about a possible origin of wave (d)

Typos: The text of Figure 7: Latitude-time. . .. should be Longitude-time. . .. Done

Please also note the supplement to this comment:
https://www.atmos-chem-phys-discuss.net/acp-2017-1051/acp-2017-1051-AC1-supplement.pdf

———————————————

[Figure]

**Fig. 1.** Latitude-Altitude cross-section of the amplitude of the SPW 1 and 2 in Period I and II. The amplitudes are calculated using GPH data from MLS.

**Period I**
**21/12/2015 - 05/01/2016**

**Period II**
**05/01/2016 - 20/01/2016**

SPW1

SPW 2

**Fig. 2.** Same as Figure 1 but estimated from MLS temperature data

**Supplement:**

[Figure]

Figure S 1: Latitude-altitude cross-section of the SPW 1 and 2 derived from MLS GPH data. The amplitude as it occurred in the Arctic winter 2015/16 is color coded, the deviation from the 12-year mean is presented in contour lines for the respective time period from MLS GPH data. The first half of the time period investigated here is called hereafter Period I and the second half Period II.

[Figure]

Figure S 2: Vertical profile of the zonal mean zonal geostrophic wind averaged between 50 and 60°N for Period I (top) and II (bottom). Data are derived from MLS GPH data.

[Figure]

Figure S 3: Top: Longitude-Altitude cross-section of the absolute GW drag in Period I. Bottom: Longitude-altitude cross-section of the geostrophic zonal wind amplitude (colored contour) and the absolute GW drag amplitude in m/s/d (contour lines) in Period I. The plot shows the deviation from the zonal mean of the wavenumber 1 filtered atmospheric wind component and GW drag. Data are derived from SABER. Note that this is the northernmost latitude band available from SABER in Period I.

[Figure]

Figure S 4: Top: Longitude-Altitude cross-section of the absolute GW drag in Period II. Middle and bottom: Longitude-altitude cross-section of the geostrophic zonal wind amplitude (colored contour) and the absolute GW drag amplitude in m/s/d (contour lines) in Period II. The plots show the deviation from the zonal mean of the wavenumber 1 and 2 filtered atmospheric wind component and GW drag. Data are derived from SABER.

---

## Author Comment (AC2) · 4 Mar 2018

Dear Referee #2, We are grateful for you friendly and constructive review. Based on your comments and suggestions the manuscript is now improved. In the following point-by-point responses the reviewer comments are in italics, our responses are in blue.

The only issue I have is that I am not convinced that the authors showed sufficient evidence that wavenumber 2 is generated by in situ gravity wave drag at the high latitudes. As we wrote in the paper, we are not able to proof that the polar mesospheric SPWs Period II and especially in Period I are generated by in situ GWD at the high latitudes

since we have the absolute GWD only and some other limitations. However our results support the assumption that especially the SPW 2 is in situ generated by longitudinally filtered GWD. We strengthen the emphasis in the appropriate text passages and in the conclusions that our results "only" support the assumption of in situ generation by GWs and instabilities and not prove it.

Page 1, line 15: change to ". . .show that all three mechanisms. . ." Done

Page 1, line 24: change to "In addition to these global anomalies. . ." Done

Page 3, line 8: change to "longitudinally variably" or "variably in longitude" Done

Page 3, line 10: change the word order to ". . .how favorable the conditions are in the. . ." Done

Page 3, line 11: awkward word order, change to ". . . in situ generation of the quasi 2-day wave, for example." Done

Page 4, line 3-4: change "in all latitudes. . ." to "at all latitudes" Done

Page 4, line 14: should be "atmospheric parameters"? Done

Page 6, Figure 2: Yes the PNJ is much stronger than the 12-year mean, but is this surprising? Doesn't the NH PNJ move around a lot from year to year, so that the 12-year mean is of course a little washed out? It would make a stronger case that this PNJ is exceptional if the figure showed the spread of all 12 years instead of just the mean. So for example, a line plot of the zonal mean zonal wind averaged between 50 and 60 N and 40 and 60 km versus time for the entire Period I.

The reviewer is right, the mean of the PNJ has a large standard deviation in the northern hemisphere as shown left in the zonal mean zonal wind profiles averaged between 50 and 60°N for both periods. However, for our explanations of the vertical and horizontal propagation of the SPWs into the subtropical mesosphere, we need the latitude-altitude cross-section of the zonal wind. We added a figure of the zonal mean zonal

wind vertical profile averaged between 50 and 60°N for Period I and II in the Supplements and added an appropriate text phrase on page 6 line 7/8 and on page 8 line 16/17.

Page 7, line 17: change word order to "used here" Done

Page 7, line 21: change "shift in vertical" to "shift in the vertical" or "vertical shift" Done

Page 8, line 24 and 26: change to "in the vertical" Done

Page 10, section 5 heading: change to "Why does the SPW 2 dominate in Period II?" Done

Page 11, line 12: change to ". . .only possible in a weak zonal mean zonal wind." Done

Page 14, Figure 7 caption: The date range of the SABER data should be added to the caption. Is it the entire Period II? This was not clear in the text either. Done

Page 14, Figure 7: I'm not convinced that it makes sense to filter the GW drag for wavenumber 2. If wavenumber 2 is the dominant wavenumber in the zonal wind, and if the GW drag is the cause of this, then the unfiltered GWD should show that. It might be more interesting to see what the unfiltered GWD looks like, because the wind doesn't just feel the wavenumber 2 GWD. The numbers are also very small for the GWD. On the order of 0.5 to maybe 2 m/s/d it looks like, an order of magnitude smaller than those from Smith 2003 for example. Again, it might be more fair to show the GWD from all wavenumbers.

The reviewer might be right that showing the GWD from all wavenumbers is fairer to the reader. However, we think that it is confusing to show all wavenumbers when looking for the origin of wavenumber 2 only. We decided to add two figures to the supplements (see below) including the unfiltered absolute GWD and, similar to Figure 7 in the paper, the filtered zonal wind and absolute GWD for wavenumber 1 in Period I and for wavenumber 1 and 2 in Period II. We hope that this satisfies the reviewer.

This figure shows that there is a non-uniform GW drag between 40° and 50 °N in Period I, the northernmost latitude band available in Period I due to the southern yaw cycle of SABER in that time period. The geostrophic winds from MLS in polar latitudes give a hint, that there could be an in situ generation by GWs between 60° and 70°N and an additional in situ forcing by instabilities between 50° and 60°N. This is supported by non-uniform GWD at 40°-50°N but as we already said: It is not a proof.

In Period II the GW drag filtered for wavenumber 2 is stronger and in a more robust phase relation with the zonal wind compared to the wavenumber 1 filtered GW drag and zonal wind in 60° to 70°N. The amplitude of the GW drag is indeed much lower in our case compared to Smith et al. 2003 but so is the amplitude of the SPW. Additionally SABER "sees" only a part of the GW spectrum. For a sensitivity function see Ern et al. (2018), Fig. 3d. Further, there is a low bias on the observable GW drag (cf. Ern et al., 2004, 2011). We added an appropriate discussion on page 16 line 20 - 23.

Page 15, line 18: change "primary" to "primarily" Done

Page 16, line 14: change to "raises the question" Done

Please also note the supplement to this comment:
https://www.atmos-chem-phys-discuss.net/acp-2017-1051/acp-2017-1051-AC2-supplement.pdf

───────────────────

Period I

Period II

**Fig. 1.** Profile of the geostrophic zonal wind averaged between 50 and 60°N. Data are derived by MLS GPH data.

**40° - 50°N**

Fig. 2. Abolute GW drag averaged between 40 and 50°N in Period I. Data are derived fom SABER data.

[Figure]

50° - 60°N          60° - 70°N

wave 1

wave 2

**Fig. 3.** Abolute GW drag in Period II filtered for different wavenumbers. Data are derived fom SABER data.

---

## Author Comment (AC3) · 4 Mar 2018

Dear Referee #3, We are grateful for you friendly and constructive review. Based on your comments and suggestions the manuscript is now improved. In the following point-by-point responses the reviewer comments are in italics, our responses are in blue.

I have two particular requests for changes before the paper can be accepted for publication. My first request is that the authors expand on their explanations of the SABER measurements. In particular, there should be a more expansive description of the technique and its limitations. At present the measurements are simply described as being

of "absolute GW momentum fluxes" and the reader is provided with references for more substantive explanation of the technique. However, the SABER measurements are a central part of the paper and there should be i) a paragraph of explanation describing the technique at the point where it is introduced and ii) some discussion of the limitations of the technique.

In respect of the latter, I believe that these GW momentum flux measurements actually yield a lower bound rather than a fully-constrained value, since the estimates of horizontal wavelength depend on the angle between the satellite's orbit and the phase fronts of particular gravity waves.

The reviewer is right. The limitation of the GW drag calculation using SABER data should be mentioned and discussed in the paper. The limitation of the GW drag calculation using SABER data are that SABER "sees" only a part of the GW spectrum, namely the inertia GWs (the larger ones) and the known low bias of the observable GW drag as discussed in Ern (2004). We added appropriate text passages in the "Instruments and methods" part on page 6 line 3-6 and in section 6 on page 16 line 20 -23 where the results are discussed. We also expanded our explanation of the GW drag calculation from the SABER measurements in section 2 starting on page 5 line 27 and ending on page 6 line 13 .

The arguments on p14 in paragraph 2 about the longitude-altitude cross section of wavenumber 1 filtered winds and the non-uniform GW drag at 50N with a wavenumber 1 structure refers to figures "not shown" – these figures would confirm the arguments being made by the authors so they should be included.

We included a figure on the wavenumber 1 filtered zonal wind from MLS for the polar latitudes and for the zonal wind and GW drag from SABER for $40°$-$50°$N to strengthen our arguments. Additionally we put a figure of the unfiltered longitudinally non-uniform GW drag at $40°$-$50°$N into the Supplements (see Fig. S3).

The arguments about the wavenumber 2 component of gravity-wave drag associated

with Fig 7 would be strengthened by some explanation of the total drag and its other component wavenumbers. There does seem to be a wavenumber 2 component as shown, but how big is it compared to the zonal-mean value and the other wavenumbers?

Since reviewer #2 has a very similar request we added the non-filtered GW drag for the different latitude band as well as the, for wavenumber 1 and 2, filtered version in the Supplements (see Fig. S4). In Period II the GW drag filtered for wavenumber 2 is stronger and in a more robust phase relation with the zonal wind compared to the wavenumber 1 filtered GW drag and zonal wind in $60°$ to $70°$N. We added a comment on this issue on page 16 line 11-13.

My second request is that some of the figure be made larger. As presented, some of the contours in Figs 2c, 3c, 4 and 7 are very faint, hard to read and/or close together. I think that just making the figures larger would solve this problem. Done

MINOR POINTS P1, l15, suggest ". . . show that all three mechanisms. . ." Done

Figure 1 caption, suggest ". . .from MLS temperature. . ." Done

P3, l21, the final sentence "The vertical propagation of. . . (Lin, 1982)" would make much more sense if put at the start of that paragraph. Done

P4, l17, should be ". . .winds are needed". Done

P4, l20, should this be aËĘ(-1) @/(@â′LEËĞ ) ? Done

P5, l5, suggest ". . .which results in a westerly wind. . ." Done

P5, l9, suggest ". . .and following it up into. . ." Done

P5, l14, suggest ". . .TIMED satellite and measures temperatures. . ." Done

P5, l16, suggest ". . .geometries about every 60 days. For the period of. . ." Done

P8, l2, suggest ". . .Period I, as is the areas of. . ." Done

P11, Fig 5, what causes the missing data at days -2 to 0? The missing data are caused by a data gap in the MLS raw data which is not visible in the other figures due to averaging over the 15 days period.

P14, l3, suggest ". . .not able to investigate whether wave (d). . ." Done

Please also note the supplement to this comment:
https://www.atmos-chem-phys-discuss.net/acp-2017-1051/acp-2017-1051-AC3-supplement.pdf
* * *